**Data Availability Statement:** Data can be accessed through the Measure DHS website (https://dhsprogram.com/) as the study is based on a secondary analysis of BDHS data. The working

# How do traditional media access and mobile phone use affect maternal healthcare service use in Bangladesh? Moderated mediation effects of socioeconomic factors

**Md Ruhul Kabir** [1,2] *

1 School of Communication and Film, Hong Kong Baptist University, Kowloon Tong, Hong Kong,
2 Department of Food Technology & Nutrition Science, Noakhali Science & Technology University, Noakhali, Bangladesh

* 20481713@life.hkbu.edu.hk, ruhul109@gmail.com

## Abstract

### Background

Exposure to traditional media (TV, radio, and newspapers) and the use of mobile as an interpersonal communication tool allow for a variety of information provision. The purpose of this study is to investigate how women's media and mobile access affect maternal health service (MHS) utilization. The study also aims to look into the moderated mediation effects of socioeconomic variables on the association mentioned above.

### Methods

The study analyzed reproductive and media data of 5,011 ever-married women extracted from the latest nationally representative Bangladesh Demographic and Health Survey. Hierarchical logistic regression and moderated mediation analysis are performed to determine the association.

### Results

Only 26.9% of women used mobile for health service use, while more than 55% had media access. Media access is significantly associated with all three types of MHS use; mobile usage also has a significant association with antenatal and delivery care. When women have both access to media and mobile, the likelihood of delivering in a health facility increased by 1.82 times (AOR: 1.82, 95%CI: 1.51, 2.20) which is slightly better than having access to only one type of media channel. Women's education, household wealth, place of residence, religion, and current working status are among the socioeconomic factors associated with access to media and mobile. Women's education mediates the relationship of media and MHS; however, the mediation effect of women (β: .45; LLCI: .21, ULCI: .68) on the association of media and place of delivery is seen to be moderated by household wealth. Women who belong to well-off families moderates positively (Effect: .33, 95%CI: .27, .40) the education effect of media and where to deliver. Place of residence, another moderator,

data file is attached as a supplementary file (S1 File_Working Data Set).

**Funding:** The author(s) received no specific funding for this work.

**Competing interests:** The authors have declared that no competing interests exist.

significantly moderates (Effect: .09, BootLLCI: .02, BootULCI: .16) the mediation effect of women's education on the association of media and antenatal care visits; women living in urban areas seems to have positively moderated the education effects on the mentioned association.

## Conclusions

Provision of media access and mobile use indicate better utilization of MHS in Bangladesh, and women's education mediates these relationships via the influence of household wealth and area of residence. Therefore, while planning interventions to increase MHS use, its relationships with the media and mobile use should be extrapolated. The collective use of these channels could be a catalyst for the success of health promotion initiatives to improve women's health behaviors, build community capacity, and create mass awareness that supports the optimal use of MHS in Bangladesh.

## Introduction

Studies on maternity care in developing countries like Bangladesh are constantly being looked at by academics who are concerned about the low usage of essential maternal health services [1, 2]. Despite mounting pressure to reduce it, the maternal mortality ratio (MMR) continues to be a major concern for many countries and remains at the top of the list of global health priorities. According to the world health organization (WHO), identifying and addressing systemic and social barriers to optimal maternal health care should be a priority going forward. WHO also acknowledges that inequalities in healthcare access, including wealth disparities, geographical disparities, and lack of skilled birth attendants, account for roughly 810 maternal deaths per day globally, and tragically, 94% of these deaths occur in areas with poor-resource settings [3]. Insufficient health education [4, 5] and lack of information [3] are major considerations that repeatedly resonate when deciding to use maternal health care. For this reason, it is important to consider how to inform women about their own healthcare better and increase the idea that they should use health services more frequently in low- and medium-income settings, given the lack of awareness that comes from women's poor education, socioeconomic disparities, [1, 6] and the systematic failure of health systems to meet health service expectations [7, 8]. In that manner, they may be able to grasp the severity of the problem, appraise the risks, and realize their own health rights to ensure that the pregnancy outcomes are not harmed [9].

The relationship between traditional media and health services remains an important field of research, even in the era of digital capitulation. As women's education, household wealth, and residence are persistent among all the socioeconomic factors on the use of maternal health services [2], it's reasonable to assume that not all women will reap the benefits of living in the digital world. The importance of conventional media sources (TV, radio, and newspapers) as a conduit of various information distribution, including health service promotion contents, cannot be overstated since the mass media may be the most important source of information for many people [10]. Access to traditional media sources can empower women, even if only passively, by offering health information in a variety of formats, such as audio-visuals (advertisements), role-play dramas, case studies, written messages (announcements), songs, and so on. Although maternal education level and other social factors, norms, and beliefs do have a role in maternal health service information acquisition [11], mass media, on the other hand, has the power to reach a considerable audience with health promotion and awareness messages

[12]. For instance, people can be informed about vaccinations, important dates, screening locations, and how the entire process of using healthcare services works; in general, media can promote social action by connecting members of the community [12]. The use of maternal health services (MHS) should be promoted and backed up by the families and community people due to the patriarchal societal nature [13], and mass media can play an essential role in encouraging the community to develop capacities around the complete pregnancy care. One study in South Asia revealed that maternal health service utilization is higher among women exposed to mass media across countries [14]. However, the effect of mass media on the use of maternal health services is not straightforward. Understanding the nature of their association is vital in developing any context-specific intervention plan.

Furthermore, employing a mobile phone as a tool for interpersonal communication is an essential component of healthcare communication because its applications are expanding over time, and the digitalization effect should reach the limits as time passes. Additionally, women's exposure to mass media is aided by their availability to mobile phones and constant connectivity with health practitioners [15]. Mobile technology, as a faster mode of communication, has the potential to significantly improve access to emergency obstetric care (EmOC), which includes care necessary during and immediately following delivery [16]. The use of mobile phones to access maternal health services has enormous promise because it can help reduce wait times for medical care, connect women with health providers, and give critical emergency consultations [17]. A study in Bangladesh found that mothers who own mobile phones and are exposed to mass media are more likely to use prenatal and delivery care. The study concluded that enhanced media and mobile access could help promote maternal health access [18]. However, the potential of mobile phones on the improvement plan of maternal health services in LMICs needs to be explored further [16].

This study, however, extends the examination of the relationship between mobile phone use and media access and maternal health service utilization while considering the potential sociodemographic-economic factors that might mediate and moderate the relationship. The rationale of the study is to assess the influence of mobile phones as a mode of interpersonal communication and media access for information provision on the receipt of critical maternal health care in Bangladesh, specifically antenatal, delivery, and contraceptive use. As discussed above, using mobile phones for seeking maternal health services and being exposed to mass media on a frequent basis can act as catalysts in utilizing MHS in a country like Bangladesh. Therefore, understanding the dynamics of the connection may aid in the development of any intervention strategy related to it. The involvement of important socioeconomic and demographic factors will also be examined in this study in order to measure and assess their strength of association.

The selection of the maternal health services is motivated by the WHO's assertion that the risk of maternal death can be reduced by increased access to modern contraception and improved access to high-quality care prior to, during, and after childbirth [19]. The study also hypothesizes that using more channels of health information seeking might increase the likelihood of receiving these health services more often. According to the channel complementarity theory (CCT), motivated individuals are inclined to acquire useful information from all the available sources perceived to fulfill the information need. The theory postulates the use of multiple sources during information seeking about a particular issue to make a useful difference in information acquisition [20].

## Maternal health service use and effects of socioeconomic factors

Several socioeconomic-demographic characteristics were found to be significant in the research of maternal health service utilization, with the educational attainment of women and

household wealth status topping the list. Apart from that, rural-urban residency, women's age, and religion all have an effect on how much antenatal, delivery and postnatal care is used [1, 21]. To better understand women's socioeconomic vulnerability in relation to health service usage, this study examines the mediating function of women's education in the link between media, mobile access, and MHS use. As media exposure has a beneficial effect on women's education and subsequent empowerment [22], knowing the function of education in this process may provide useful information. Is it possible for other socioeconomic characteristics to further moderate the mediating role of women's education? The study also seeks an answer to that.

## Methods

### Study design

The study was based on the secondary analysis of the recent 2017 Bangladesh Demographic and Health Survey (BDHS) data accessed through the Measure DHS website (https://dhsprogram.com/). The study generated a nationally representative sample and was conducted on ever-married women aged 15–49 years old to produce periodic data on the community's socio-demographic and health status, mostly women and their children. The latest data was the eighth national survey which used previously formed enumeration areas consisting of around 120 households to create the sampling frame. The collection of samples was based on two-staged stratified sampling, and more about sampling and study design can be found elsewhere (https://dhsprogram.com/methodology/survey/survey-display-536.cfm). The study used a women's questionnaire, which includes women's reproductive and contraceptive history (family planning), socio-demographic background, healthcare service use, media access, mobile technology, etc. For media access, how frequently women read a newspaper, watch TV, or listen to radio were asked; and for mobile use, whether women used mobile for health service was asked (for instance, consultation, etc.). Data were collected through face-to-face interviews by well-trained staff with a validated questionnaire. After careful consideration of data availability on media access and mobile usage, a total sample of 5,011 ever-married women who delivered at least one child in the last three years preceding the survey is included in this study (S1 File).

**Dependent and independent variables.** The effect of media access and mobile usage was tested against the three important maternal health service use, namely antenatal care visits, delivery place, and contraception. The categorization of three types of health services and other exogenous variables is presented in Table 1. The main exogenous variables of this study were mobile usage for receiving health services and access to media.

Mobile usage for health service use: Women were considered to have utilized mobile for health service use when they have either contacted healthcare providers for consultation, to know their whereabouts, including their chamber time, or any other reasons that might involve health service use in their last pregnancy and child delivery period. The use of mobile for health services facilitates communication with health providers and is considered an interpersonal communication channel (participant's use of mobile for health services is presented in Table 1). The type of mobile phone (smart or feature) used by the women were not specified in the dataset; hence, we consider it was at least a feature phone. Moreover, the dataset does not include the level of internet use, so participants' use of mobile via the internet is unknown.

Traditional media access: Women were considered to have media access when they read newspapers, watched television, or listened to radio at least once a week in their last pregnancy period. Women were considered to have limited access when they had no exposure at all or had exposure less than once a week. As participants' level of internet use is unknown, their

**Table 1. Description of exogenous and outcome variables.**

| Exogenous variables | Variables and categories |
|---|---|
| Individual and household-level factors | • Women's age (15–19 years, 20–29 years, 30 or more) |
|  | • Education (Primary/No education, secondary, higher) |
|  | • Currently working (Yes, no) |
|  | • Religion (Islam, others) |
|  | • Household wealth (Poorest, poorer, middle, richer, richest) |
|  | • Health decision making (Wife alone, wife and husband together, husband or others) |
|  | • Birth order (1st, 2nd, 3rd, or more) |
| Locality factors | • Area of residence (Rural, urban) (used as a proxy for distance to health facilities) |
|  | • Administrative divisions |
| Mobile and media access factors | • Use of mobile (feature phone) for medical services (Yes, no)[1] |
|  | • Media access (frequency of reading newspapers/ listening to the radio/ watching television) |
|  | Have some access: Watched TV/listened to radio/read newspapers at least once a week |
|  | Limited access: Watched TV/listened to radio/read newspapers less than once a week or not at all. |
| **Outcome variables** |  |
| Antenatal care | Number of antenatal care visits for the last pregnancy (<4 visits, $\geq$ 4 visits) [a] |
| Place of delivery | Place where last delivery has taken place (Health facility, home/other places) [b] |
| Use of contraception | Uses of any kinds of family planning/contraception methods at the time of the survey (Using, not using) |
| Specific Reasons | Frequency (%) |
| To ask what to do (consultation) | 797 (46.6) |
| To contact the service provider (whereabouts/appointments) | 660 (38.6) |
| Other reasons (Transportations, medical arrangements, etc.) | 251 (14.8) |

[a]WHO recommends at least four visits for ANC;

[b]Equipped health facility delivery by skilled professionals might facilitate better child delivery and can manage complications well [4, 25, 26].

[1]Use of mobile for health services.

usage of social media is not considered for this study. The other exogenous factors that were selected to control include individual, household, and locality level factors (Table 1).

Among the three dependent variables, ANC visits were categorized into whether women had visited recommended number ($\geq$4) of times or not to any kinds of health providers. Place of delivery includes whether women had delivered their last child in a health facility or not. Use of contraception consists of the contraceptive method (any type) women adopted or not. These three types of maternal health services were chosen to be studied based on the importance of maternal health outcomes.

**Use of channel complementarity theory (CCT) as a theoretical skeleton.** The present study applies CCT to assess how traditional media as one channel and mobile as an interpersonal communication channel can affect maternal health service use. The theoretical approach posits that media consumers may use complementary channels while seeking information for a particular topic. Dutta-Bergman (2004) asserts that information received from one source may drive users to seek other sources of information and lead to interpersonal interaction with others [20], implying that communication begets communication. Health communication research shows that users tend to learn from various media sources, including interpersonal channels and what they have learned from their physicians. Research in the health communication domain also supports the CCT approach that information seekers find a way to use interpersonal and mass media information in complementary ways [23]. Thus, this study seeks to identify the influence of the complementarity effect of mobile, which women have

primarily used for interpersonal communication with their physicians, and traditional media on their overall maternal healthcare service utilization.

**The definition of mediation and moderation variables.** Mediators and moderators are variables that have an effect on the relationship between two independent variables. Mediators shed light on how or why two variables are inextricably linked (the process). On the other hand, moderators have an effect on the strength and direction of that association [24].

## Data analysis

Descriptive statistics are presented using frequencies and percentages. A Chi-square test is performed to assess the relationship between the predictors and the outcome variables. In the first regression analysis, multivariate logistic regression is conducted to analyze the effect of socio-demographic predictors on the use of mobile phones and media access variables. In the second stage, hierarchical logistic regression is performed to find out whether introducing the main exogenous variables (media access and mobile use) in the regression model improves the overall fit or not. The combined effect of the main exogenous variables is also presented. Adjusted Odds ratios (AOR's) with 95% confidence intervals (95% CI) are presented to assess the strength of associations among the categories of different variables and their subsequent influence on the dependent variables. SPSS 26.0 software facilitates the analysis process; $p < 0.05$ is considered statistically significant.

The AOR of individual, household, and locality level factors is not presented in the second regression analysis since our main concern is investigating media access and mobile usage's contribution to essential maternal medical services. According to channel complementarity theory, the use of maternal health services is regressed against media access, mobile usage, and for women who had both the access (media and mobile). Women who had the advantages of having both the media and mobile access are regressed to explore their likelihood of using essential health services to women who did not have access to either of them.

For moderated-mediation (MM) models, bias-corrected 95% confidence intervals from 5,000 bootstrap random samples are used by using the Hayes SPSS PROCESS macro version 4.0, a software package to analyze moderated-mediation. If zero is not contained within the confidence intervals (CI) computed by the bootstrapping procedure, one can conclude that the indirect effect is indeed significantly different from zero at $P < .05$. The macro produces a *z* statistic, a sampling distribution against which the moderator and mediator are tested. Direct effect, interaction effect, and conditional indirect effects are produced by the analysis. To further illustrate the moderating effects, sample slopes analysis is presented with focal points at the mean, and one standard deviation high and low (± 1Sd) of the moderators (only high and low SD for two categories) and mediator variable to determine if the slopes of the regression equations for high and low values of the interaction are different from zero or not.

A two-sept analysis was conducted to test the significance of the moderated mediation models; in step one, women's education is tested against all the dependent variables for their mediation effect, and in step two, the other selected exogenous variables are tested to assess whether other variables moderate its effect or not. The mediation (indirect) effect is tested by using Model 7 of PROCESS macro, and for moderated-mediation effect (conditional indirect effect), Model 14 is used. All the variables included are dummy coded to facilitate the analysis process. First, the relationship between media and dependent variables is explored via the moderated-mediation model, and then the effect of mobile use is investigated. In the moderated-mediation models, the mediation effect of women's education on the association of media access and maternal health service use variable is presented along with the moderation

**Table 2. Frequency and percentage of study variables.**

| Variables | Number (%) | Variables | Number (%) |
|---|---|---|---|
| **Mobile and media variables** | | **Health service variables** | |
| Own a mobile phone | | Number of ANC visits | |
| Yes | 3077 (61.4) | Four or more visits | 2415 (48.2) |
| No | 1934 (38.6) | <4 visits | 2596 (51.8) |
| Used mobile phone to get health service/advice | | Place of delivery | |
| Yes | 1348 (26.9) | Health facility | 2520 (50.3) |
| No | 3663 (73.1) | Home/other places | 2491 (49.7) |
| Access to media | | Use of contraception | |
| Have some access* | 2771 (55.3) | Using | 3357 (67.0) |
| Limited access | 2240 (44.7) | Not using | 1654 (33.0) |

(Total sample: 5011;

*Watched TV/listened to radio/read newspapers at least once a week).

effect of another variable. Mobile use also affects the MHS service the same way as the media access; hence, the result of the mobile use is not presented.

**Ethical considerations.** The study is based on the secondary analysis of BDHS data; hence, no ethical permission is required. The data is accessed by submitting a request to the Measure DHS website.

## Results

Table 2 depicts the frequency and percentages of study variables representing participants' health service use, media, and mobile access. More than 61% of women owned a mobile phone; however, only 26.9% of them used it for health services. More than half of the women (55.3%) had some access to media on a weekly basis. Around half of the women did not visit a recommended number of times for ANC and were delivered at home or other places instead of health facilities. The weighted prevalence of four or more ANC visits was 47%, health facility delivery was 49.9%, and women's use of contraception was 66.2%.

Table 3 presents the percentage distribution of outcome variables in regard to the main exogenous variables. Media access was significantly associated with all the maternal health-related variables. The association remained significant when media access was added to the mobile usage variable. Mobile use was not found to associate with the use of contraception. 67.7% of women made recommended number of ANC visits when they had both media and mobile access. More than 70% of women delivered in health facilities used different types of contraception when exposed to media and used mobile.

Women and their husbands' education and their current working status were significantly associated with mobile use for health services. Highly educated women had a 1.84 (AOR: 1.84, 95% CI: 1.44, 2.34) times higher chance of using mobiles for health services than women who either did not complete primary education or did not study at all. Women used mobile for health services more often while having their first child (AOR: 1.47, 95% CI: 1.19, 1.78). Women's education is also significantly associated with media access, so does household wealth, religion, and area of residence. Women who belonged to the richest household were 32.86 (AOR: 32.86, 95% CI: 24.01, 44.05) times more likely to be exposed to media than their poorest counterparts. For women who had a religion other than Islam, their chances of having media access increased by 1.69 times (AOR: 1.69, 95% CI: 1.31, 2.17) (Table 4).

**Table 3. Percentage distribution of outcome variables in response to main exogenous variables.**

| Variables | Number of ANC visits | | Place of delivery | | Use of contraception | |
|---|---|---|---|---|---|---|
| | ≥ 4 visits | p-value | Health facility | p-value | Using contraception | p-value |
| Mobile use for health service | | | | <0.01** | | <0.37 |
| Yes | 60.2 | <0.01** | 63.5 | | 68.0 | |
| No | 43.8 | | 45.4 | | 66.7 | |
| Media access | | | | <0.01** | | <0.05* |
| Limited access | 34.9 | <0.01** | 35.2 | | 64.3 | |
| Have some access | 59.0 | | 62.5 | | 69.2 | |
| Both media and mobile access | | | | <0.01** | | <0.03* |
| Yes | 67.7 | <0.01** | 74.1 | | 70.1 | |
| No | 44.3 | | 45.5 | | 66.4 | |

P-value derived from chi-square test;

**Significant at <0.01 level,

*Significant at <0.05 level. Percentages represent values within groups of exogenous variables.

After controlling for individual, household, and locality level factors in blocks 1 and 2, the results of the hierarchical logistic regression analysis from the final block on the association between the use of mobile phones and media access with different maternal healthcare variables are shown in Table 5. The findings show that women who utilized mobile phones for health care services had significantly greater odds (adjusted) of reaching an acceptable number (four visits) of ANC visits (AOR 1.48, 95% CI: 1.26, 1.67), and health facility delivery (AOR 1.68, 95% CI: 1.45, 1.94) than women who did not. Women with some media accesses had 1.50 (AOR: 1.50, 95% CI: 1.30, 1.73), 1.33 (AOR: 1.33, 95% CI: 1.50, 1.54), and 1.24 (AOR: 1.24, 95% CI: 1.07, 1.43) times higher odds of achieving an adequate number (≥ 4 visits) of ANC visits, delivering at the health facility and using contraception than the women with limited media access. The odds of receiving ANC care, delivering in health facilities, and the use of contraception increased significantly when women had both media and mobile access. For instance, the odds of delivering a child in a health facility increased by 1.82-fold (AOR: 1.82, 95% CI: 1.51, 2.20) when women had media access and used mobile for seeking medical health services.

## Moderated mediation effects

The women's education mediates the relationship of media access and place of delivery (Coeff: .045, p<0.01) (Fig 1), and the association is further moderated by the household wealth (Women education*household wealth interaction effect: .09, P<0.01) (Index of moderated mediation: .03, 95% CI: .01, .06). The effect of media access on delivery is higher when women have higher education and belong to affluent households. The conditional indirect effect of media access on the place of delivery at different levels of household wealth entails that media access affects better with increased women education and when women belong to a rich household (Effect: .33, 95% CI: .27, .40) (Fig 2) (Table 6).

In the relationship between media access and ANC visits, women's education positively mediates the association significantly (Coff: .67, p<0.01) (Fig 3). However, the relationship is significantly moderated by the place of residence (Women education*Place of residence interaction: .22, P<0.02) (Table 7). The index of moderated mediation (Effect: .09, 95% CI: .02, .16) confirms the moderated effect. The conditional indirect effect of the media access on the ANC

**Table 4. Factors affecting media access and mobile use.**

| Variables | Mobile use for health service | Media access |
|---|---|---|
| | AOR | AOR |
| Women's education level | | |
| Higher | 1.84 (1.44, 2.34) ** | 1.54 (1.12, 2.01) ** |
| Secondary | 1.29 (1.09, 1.53) * | 1.22 (1.04, 1.44) * |
| Primary/No education | 1 | 1 |
| Husband's education level | | |
| Higher | 1.88 (1.42, 2.53) ** | 1.13 (.85, 1.51) |
| Secondary | 1.55 (1.21, 2.03) ** | 1.16 (.95, 1.52) |
| Primary/No education | 1 | 1 |
| Household wealth | | |
| Richest | 1.31 (1.02, 1.68) * | 32.86 (24.01, 44.05) ** |
| Richer | 1.09 (.88, 1.39) | 12.70 (10.02, 16.25) ** |
| Middle | 1.19 (.96, 1.50) | 8.16 (6.50, 10.02) ** |
| Poorer | 0.95 (.77, 1.20) | 4.01 (3.20, 4.09) ** |
| Poor | 1 | 1 |
| Birth order | | |
| 1st child | 1.47 (1.19, 1.78) ** | 1.49 (1.21, 1.80) ** |
| 2nd child | 1.13 (.95, 1.34) | 1.49 (1.28, 1.86) ** |
| 3rd or more | 1 | 1 |
| Current working status | | |
| Yes | 1.35 (1.19, 1.56) ** | 1.13 (.98, 1.31) |
| No | 1 | 1 |
| Religion | | |
| Others | .91 (.72, 1.16) | 1.69 (1.31, 2.17) ** |
| Islam | 1 | 1 |
| Area of residence | | |
| Urban | 1.04 (.89, 1.21) | 1.31 (1.10, 1.51) ** |
| Rural | 1 | 1 |

AOR: Adjusted odds ratio;

** Significant at <0.01 level,

* Significant at <0.05 level; Women's age and health decision-making abilities did not significantly affect the analyses.

visits at different levels of the place of residence entails that higher educated women with media access and living in urban areas (Effect: .33, 95% CI: .26, .40) had a higher propensity of ANC visits than their rural counterparts (Fig 4).

## Discussion

This study analyzed the effect of mobile phone use and media access, two communication channels, on maternal healthcare service utilization in Bangladesh. The moderated mediation effect of predictors on the association between media and mobile use with maternal health services is also explored. The study discovered that mobile phone use and media access are associated with the use of maternity health services. According to the findings, exposure to traditional media and a mobile phone as a medium of interpersonal connection results in increased antenatal visits, institutional births, and contraceptive use, albeit in a limited way on some occasions. Even though the ensuing effects (AOR's) are not dramatic, the findings are

**Table 5. Association of maternal health service use variables with mobile and media access factors controlling individual, household, and locality level factors.**

| Variables | Number of ANC visits | Place of delivery | Use of contraception |
|---|---|---|---|
| | AOR (95% CI) | AOR (95% CI) | AOR (95% CI) |
| **Block 1: Individual and household level factors** | | | |
| Women's age, education, current working status, religion, household wealth, health decision-making abilities | | | |
| **Block 2: Locality level factors** | | | |
| Area of residence, administrative decision | | | |
| **Block 3: Mobile use and media access** | | | |
| Usage of mobile technology | | | |
| No | 1 | 1 | 1 |
| Yes | 1.48 (1.26, 1.67) ** | 1.68 (1.45, 1.94) ** | 1.02 (0.89, 1.17) |
| Access to media | | | |
| Limited access | 1 | 1 | 1 |
| Have some access | 1.50 (1.30, 1.73) ** | 1.33 (1.50, 1.54) ** | 1.24 (1.07, 1.43) * |
| Both media and mobile access | | | |
| No | 1 | 1 | 1 |
| Yes | 1.52 (1.28, 1.82) ** | 1.82 (1.51, 2.20) ** | 1.10 (.92, 1.31) |

The odds ratio derived from hierarchical logistic regression analysis and the adjusted odds ratio (AOR) results presented here from the final block (block 3);

**significant at p<0.01 level,

*Significant at p<0.05 level; CI: Confidence interval. A statistically significant amount of variance was observed in the dependent variables in the final block after controlling the individual, household, and locality level factors in block 2 and block 3.

significant since they increase the likelihood of receiving more life-saving health care tailored to women's health. Given that Bangladesh's maternal mortality rate is perilously close to the danger level, any chance to increase maternal health service utilization should be seized with both hands. In Bangladesh, desirable maternal healthcare utilization (when women make the necessary number of ANC visits and deliver at a health facility) is still a long way off, and it is critical to make an effort to promote health service utilization concurrently with socioeconomic improvements [1].

Our findings corroborate earlier research indicating that using a mobile phone and media exposure improves maternal health outcomes. It enables pregnant women to communicate with health care practitioners and obtain information about the importance of prenatal care, childcare practices, nutritional supplementation, and immunization, among other key aspects of pregnancy and delivery care [27–29]. Mobile access can assist in maintaining

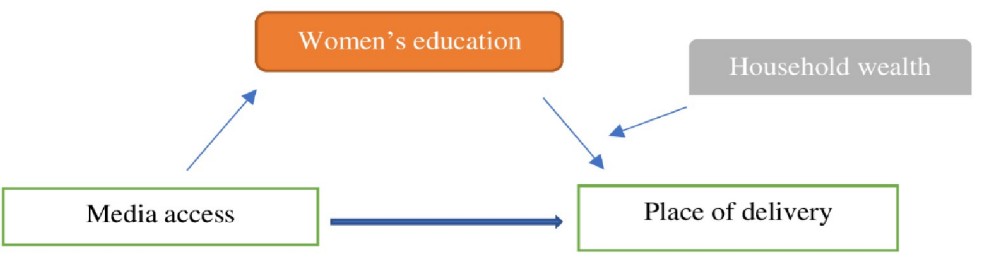

**Fig 1. Moderated mediation model.**

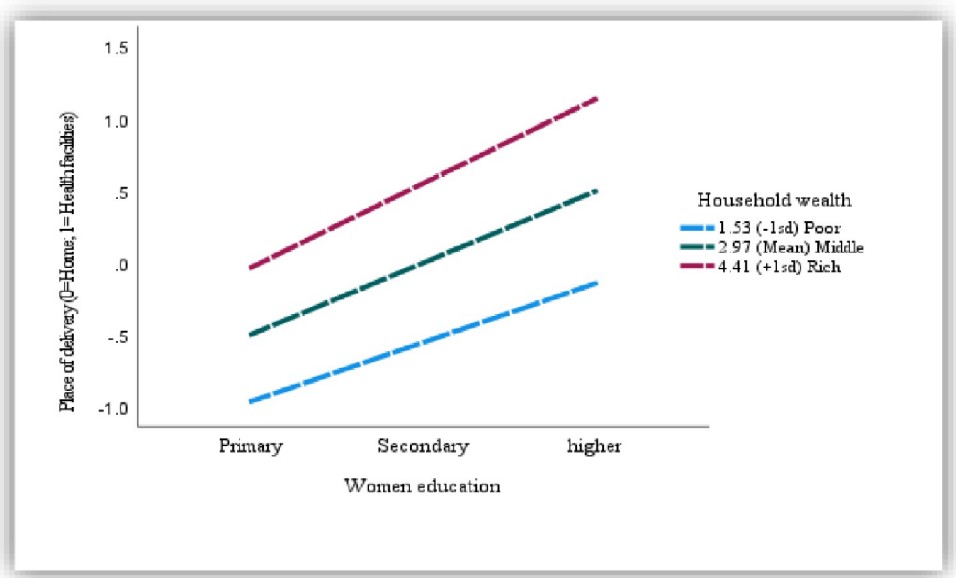

**Fig 2. Plot of the relationship (moderated mediation) between delivery place and women education at -1sd, mean, and +1sd household wealth (Conditional effect of the focal predictor).**

communication with trained health practitioners and arranging transportation in an emergency, as being late to the hospital can result in postpartum hemorrhage and associated consequences [30]. In developing countries, many women may be excluded from family decisions and may be unable to make healthcare decisions for themselves; in this case, having a cell phone enables them to inquire about nearby health services. One intervention study found that utilizing a cell phone increased antenatal care service usage by a considerable proportion and that women were increasingly accessing preventive health services [31]. Further, owning a

**Table 6. Moderated-mediation model 1 (Media women education*household wealth delivery place).**

|  | Coeff. | SE | Z | P | LLCI | ULCI |
|---|---|---|---|---|---|---|
| Constant | -1.74 | .11 | -15.95 | .00 | -1.95 | -1.52 |
| Media | .40 | .07 | 5.67 | .00 | .26 | .54 |
| Women education (Weducation) | .45 | .12 | 3.77 | .00 | .21 | .68 |
| Household wealth (Hwealth) | .31 | .04 | 7.97 | .00 | .23 | .39 |
| Weducation*Hwealth | .09 | .04 | 2.45 | .01 | .02 | .15 |
| Model summary | -2LL: 5941.85 | Df: 4 | P: .00 | Nagelkrk: .24 |  |  |
|  | Conditional indirect effect of media access on the place of delivery at different levels of household wealth | | | | | |
| Household wealth | Effect | BootSE | BootLLCI | BootULCI |  |  |
| Poor [1.53 (-1sd)] | .23 | .03 | .17 | .30 |  |  |
| Middle [2.97 (Mean)] | .28 | .02 | .24 | .33 |  |  |
| Rich [4.41 (+1sd)] | .33 | .03 | .27 | .40 |  |  |
|  | Index of moderated mediation | | | | | |
| Household wealth | .03 | .01 | .01 | .06 |  |  |

BootSE/CI: Bootstrapped standard error/confidence interval.

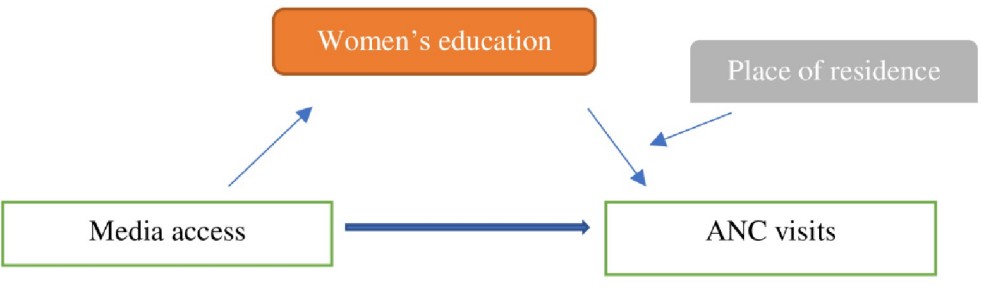

**Fig 3. Moderated mediation model.**

mobile phone is associated with a favorable attitude regarding the use of mobile technology for maternal health services [32] and can facilitate exposure to mass media [15]. Having a mobile phone boosts possible access to health information, engagement with health practitioners, and quality health services. Thus, providing women with mobile access is crucial to safeguarding their right to access and use health information [33]. The use of mobile-based maternal health information dissemination strategies may help women better comprehend and make informed decisions about their health [34]. Our study also resonates with the findings of one study that illustrates that mobile use for health services is associated with improved education, household wealth, and the current working status of women [35].

For women with access to mobile phones and mass media, we found that the number of ANC visits and delivery facilities was higher after controlling the individual, household, and locality level factors, and similar conclusions have also been found in previous studies [36–39]. Along with the mobile phone usages and media exposure, education at the individual and community level and wealthier households have positively influenced the utilization of ANC and health facility delivery [39–41]. Another study found that increased exposure to mass media has had higher odds of using ANC and health delivery facilities, complementing our study findings [39, 42]. Mass media access was also significantly associated with family planning, and other maternal health service uses [43]. Women who have had access to media have higher odds of using contraception, indicating that such women are well-informed about the

**Table 7. Moderated-mediation model 2 (Media women education*area of residence ANC care visits).**

|  | Coeff. | SE | Z | P | LLCI | ULCI |
|---|---|---|---|---|---|---|
| Constant | -1.11 | .06 | -17.95 | .00 | -1.24 | -.99 |
| Media | .67 | .06 | 10.69 | .00 | .55 | .80 |
| Women education (Weducation) | .60 | .06 | 10.64 | .00 | .49 | .71 |
| Place of residence (Presidence) | .26 | .10 | 2.56 | .01 | .06 | .46 |
| Weducation*Presidence | .22 | .09 | 2.35 | .02 | .04 | .40 |
| Model summary | -2LL: 6346.68 | Df: 4 | P: .00 | Nagelkrk: .15 |  |  |
|  | Conditional indirect effect of media access on the place of delivery at different levels of place of residence | | | | | |
| Place of residence | Effect | BootSE | BootLLCI | BootULCI |  |  |
| Rural [(-1sd) Low] | .24 | .02 | .20 | .29 |  |  |
| Urban [(+1sd) High] | .33 | .03 | .26 | .40 |  |  |
|  | Index of moderated mediation | | | | | |
| Place of residence | .09 | .04 | .02 | .16 |  |  |

BootSE/CI: Bootstrapped standard error/confidence interval.

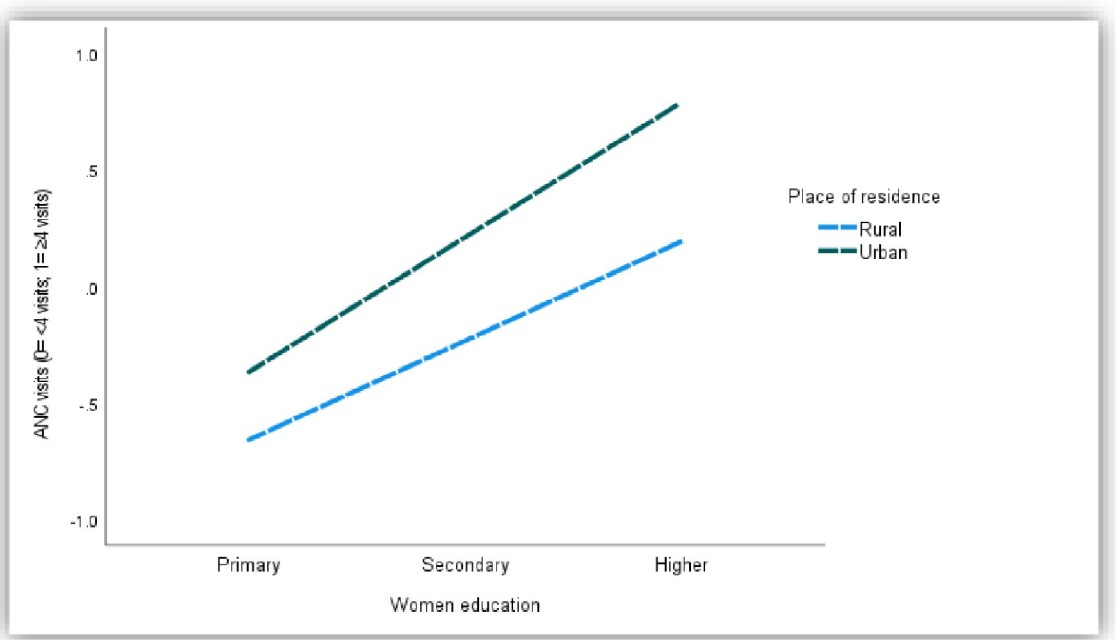

**Fig 4. Plot of the relationship (moderated mediation) between ANC and women education at -1sd and +1sd place of residence (Conditional effect of the focal predictor).**

importance of family planning and contraception use [44]. However, our results did not find any significant association between the use of contraception with the usage of mobile technology, which is not consistent with some studies [45, 46].

Without access to this health information, women may be disadvantaged or lack the understanding to make informed health decisions. That could be harmful to their quest for empowerment or health promotion. In line with the channel complementarity theory, which states that motivated individuals seek information from numerous sources, our findings show that when women used several sources, their usage of important medical services increased, though not dramatically. Women who had at least one exposure to communication (conventional media or mobile phone use) had a better chance of accessing different maternal health treatments than women who had neither access.

Focusing on the findings of moderated-mediation analysis reveals the influence of socio-economic factors on the association of media, mobile and maternal health service utilization. As the most consistent socioeconomic predictor, women's education mediates the association, which itself is moderated by (1) household wealth in the associative pathway of media access and place of delivery, (2) area of residence in the associative pathway of media access, and ANC visits. The associative nature of media access and having delivered a child in a health facility is mediated by women's improved education. The effect of education works better when women belong to rich households. Evidence from South-Asian courtiers reveals that household wealth has contributed significantly to the choice of delivery place, and mothers' education also influences handsomely [47]. Another study shows that births among women who belong to the richest group are 68% more likely to deliver in health facilities than women with the lowest wealth quintile [48]; household wealth does have a say in the relationship between media and health service use. Living in urban areas moderates women's education's

effect on the association of media and women's efforts to seek antenatal care visits. One systematic review revealed that rural dwellers are less likely to attend ANC care than their urban counterparts, and increased distance to health facilities also negatively affects the use of ANC services [48]. While developing interventions to improve women's maternal health service use, it is crucial to consider the effects of socio-environmental-economic variables that mediate and moderate the association. It entails the multifaceted nature of involvement, and evaluating approaches that involve improving social determinants of health might improve service utilization. Women's education, household wealth, and area of residence are important factors that affect maternal health service use in Bangladesh [1]. The educated-uneducated, rich-poor, and rural-urban differentials creep in everywhere when it is about maternal healthcare service use. Effective planning is imperative to reduce this gap or make an effort to keep the health service use indifferent to everybody irrespective of their education, location, and wealth. That way, it might be possible to face the low utilization of maternal services to make health services available, accessible, and affordable to all.

The main strength of this study is the generalizability criteria of the data, as it claims to be representative of the whole population. One of the main limitations of this study is the cross-sectional nature of the data, which limits the ability to determine the causal relationship between variables and their direction. Self-reported data is another limitation that can introduce respondent and recall bias (under-reporting or over-reporting). In addition, face-to-face interviews of the respondents may introduce interviewer bias; therefore, the findings should be interpreted with caution keeping in mind the limitations of controlling large data sets. Furthermore, the participants' internet access and usage, as well as their involvement with social media, were not able to be gathered in the media access section, which could have contributed additional value to this study. Improving access to media and mobile phones for receiving health services may also be greatly influenced by other cultural beliefs, practices, and norms. Therefore, it is also important to produce a thorough intervention plan that considers other relevant factors that impede the access or use of these services. Women's rights to access and use services could be improved by constantly disseminating helpful health promotion messages and digital health programs to promote women's health. Having access to media and mobile devices is not a panacea for enhancing maternal health service usage, but it could help improve interpersonal communication and provide more information to women.

## Conclusions

The study investigated the importance of having access to different kinds of traditional media and the use of mobile phones on receiving essential maternal health services in Bangladesh. Media access and mobile usage are associated with increasing use of ANC services, facility delivery, and contraception. Providing mobile and media access and spreading health messages through these mediums may assist in improving women's healthcare-seeking behavior regarding maternity care. Socioeconomic factors should also be considered, with a focus on eliminating disparities in health service utilization between urban and rural areas. The association between media and mobile components and MHS usage are mediated by maternal education, one of the most important socioeconomic determinants. The effect of improved education on the association of media and place of delivery and the number of antenatal care visits was shown to be moderated by increased household wealth and urban women living. This highlights the importance of addressing both socioeconomic and healthcare utilization concerns at the same time in order to achieve a more comprehensive outcome.

## Supporting information

**S1 File.**
(RAR)

## Author Contributions

**Conceptualization:** Md Ruhul Kabir.

**Data curation:** Md Ruhul Kabir.

**Formal analysis:** Md Ruhul Kabir.

**Funding acquisition:** Md Ruhul Kabir.

**Investigation:** Md Ruhul Kabir.

**Methodology:** Md Ruhul Kabir.

**Project administration:** Md Ruhul Kabir.

**Resources:** Md Ruhul Kabir.

**Software:** Md Ruhul Kabir.

**Supervision:** Md Ruhul Kabir.

**Validation:** Md Ruhul Kabir.

**Visualization:** Md Ruhul Kabir.

**Writing – original draft:** Md Ruhul Kabir.

**Writing – review & editing:** Md Ruhul Kabir.

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
