## [Decision Letter · Decision Letter 0]

2 Mar 2022

PONE-D-22-02833How do media access and mobile phone use affect maternal healthcare service use in Bangladesh? Moderated mediation effects of socioeconomic factorsPLOS ONE

Dear Dr. Kabir,

Thank you for submitting your manuscript to PLOS ONE. After careful consideration, we feel that it has merit but does not fully meet PLOS ONE’s publication criteria as it currently stands. Therefore, we invite you to submit a revised version of the manuscript that addresses the points raised during the review process. You are hereby  invited to use the comments and suggestions of two (2) reviewers to revise and re-submit the manuscript for consideration

We look forward to receiving your revised manuscript.

Kind regards,

Bassey E. Ebenso, Ph.D., M.P.H., M.D.,

Academic Editor

PLOS ONE

Journal Requirements:

3. Please ensure that you refer to Figures 1 and 3 in your text as, if accepted, production will need this reference to link the reader to the figure.

4. We note you have included a table to which you do not refer in the text of your manuscript. Please ensure that you refer to Tables 6 and 7 in your text; if accepted, production will need this reference to link the reader to the Table.

5.. Please include captions for your Supporting Information files at the end of your manuscript, and update any in-text citations to match accordingly. Please see our Supporting Information guidelines for more information: http://journals.plos.org/plosone/s/supporting-information. 

Reviewers' comments:

Reviewer's Responses to Questions

**Comments to the Author**

1. Is the manuscript technically sound, and do the data support the conclusions?

Reviewer #1: Yes

Reviewer #2: Yes

2. Has the statistical analysis been performed appropriately and rigorously? 

Reviewer #1: Yes

Reviewer #2: N/A

3. Have the authors made all data underlying the findings in their manuscript fully available?

Reviewer #1: No

Reviewer #2: Yes

4. Is the manuscript presented in an intelligible fashion and written in standard English?

Reviewer #1: No

Reviewer #2: Yes

5. Review Comments to the Author

Reviewer #1: I am happy reading the paper which is sound interesting and important in the context of Bangladesh using the latest round demographic dataset.

I am really satisfied reading the paper as this format. The analysis is robust and the findings are interesting although several studies already done on this.

I have few suggestions for the author, please clarify the rationality in the background section and the conclusion section is more general, please re-edit the conclusion section supporting the findings.

Weighted prevalence of the three outcomes variables are suggested.

Some grammatical errors I have found along with some blend of British and US spelling found, suggested to make it corrections carefully.

Reviewer #2: It appears that the major limitation to the study was the secondary dataset obtained from the Bangladesh Demographic Health Survey which the authors claim did not capture certain information such as the level of internet use and access by the study participants.

In an attempt to demonstrate the channel complementarity theory, the authors set out to describe the relationship between two channels; media access to health information and mobile technology use for interpersonal healthcare communication among pregnant mothers and utilization of maternal healthcare services. The paper aptly examines the secondary influence of social determinants of healthcare on these channels and consequently maternal health service utilization outcome.

However, the paper has many weaknesses that need to be addressed before it is accepted for publication:

1. Limitations in dataset analysed: It appears that the major limitation to the study was the secondary dataset obtained from the Bangladesh Demographic Health Survey which the authors claim did not capture certain information such as the level of internet use and access by the study participants. This has completely eliminated the intractable influence of social media on access to health information in a ‘computer age’. Also, in Page 12, paragraph 2, line 9 and 10, though the authors agree that using more channels of health information might increase utilization of maternal health services, it fails to demonstrate the effect of social media usage on the knowledge, attitude and practices of the study subjects.

Action: It is therefore recommended that since the data source did not capture the datasets that appropriately describe contemporary media channels such as social media and internet usage, the authors may wish to review the title of the paper to reflect the source of data.

2. Definition of key terms: While the authors defined ‘have some access’ as ‘Watched TV/listened to radio/read newspapers at least once a week’, it is completely silent on the definition of ‘limited access’. It graded the level of access to media use (some access and limited access) with accompanying degree of maternal health service utilization but failed to do same for mobile phone use assuming that ‘any kind of exposure’ to mobile technology use will positively affect maternal health service utilization outcome.

Action: The authors should clearly define ‘limited access’ so as to avoid ambiguity to the target audience.

3. Oversimplification of findings: The authors’ address a critical topic in demand creation within a complex system of factors affecting uptake of comprehensive maternal care services amongst pregnant mothers. For instance, the influence of counter moderating factors such as increased distance to health facilities on the use of ANC services were completely ignored. Also, In Page 27, paragraph 2, line 8, the authors ignored the influence of cultural beliefs, practices, and norms on the lack of women’s autonomy in making healthcare decisions for themselves in the developing world by propounding that having a cell phone to make enquiry about nearby health services is a solution.

Therefore, this paper does not sufficiently describe the balance of complex demand and supply forces for achieving desirable maternal health services outcomes. Action: Providing this information will furrther strengthen the paper

---

## [Author Response · Author response to Decision Letter 0]

8 Mar 2022

How do traditional media access and mobile phone use affect maternal healthcare service use in Bangladesh? Moderated mediation effects of socioeconomic factors

[PONE-D-22-02833]

I would like to thank the editor-in-chief, and both the reviewers for the valuable input and suggestions that I have received thus far. It was really helpful to think further to improve the manuscript. Please find the specific response from me for your comments and suggestions below. I tried to consider every suggestions that you have gracefully provided as per my knowledge and understanding.

Comments from the reviewers and response from the author:

Reviewer #1: I am happy reading the paper which is sound interesting and important in the context of Bangladesh using the latest round demographic dataset. I am really satisfied reading the paper as this format. The analysis is robust and the findings are interesting although several studies already done on this.

I have few suggestions for the author, please clarify the rationality in the background section and the conclusion section is more general, please re-edit the conclusion section supporting the findings.

Weighted prevalence of the three outcomes variables are suggested.

Some grammatical errors I have found along with some blend of British and US spelling found, suggested to make it corrections carefully.

Response from the author: Thank you for your valuable comments and suggestions to this work. As per your suggestions the rationality in the background section and conclusion section is updated. Weighted prevalence of the three outcome variables is also provided. I have gone through the manuscript again for possible language modifications. Thanks again.

Reviewer #2: It appears that the major limitation to the study was the secondary dataset obtained from the Bangladesh Demographic Health Survey which the authors claim did not capture certain information such as the level of internet use and access by the study participants.

In an attempt to demonstrate the channel complementarity theory, the authors set out to describe the relationship between two channels; media access to health information and mobile technology use for interpersonal healthcare communication among pregnant mothers and utilization of maternal healthcare services. The paper aptly examines the secondary influence of social determinants of healthcare on these channels and consequently maternal health service utilization outcome.

However, the paper has many weaknesses that need to be addressed before it is accepted for publication:

1. Limitations in dataset analysed: It appears that the major limitation to the study was the secondary dataset obtained from the Bangladesh Demographic Health Survey which the authors claim did not capture certain information such as the level of internet use and access by the study participants. This has completely eliminated the intractable influence of social media on access to health information in a ‘computer age’. Also, in Page 12, paragraph 2, line 9 and 10, though the authors agree that using more channels of health information might increase utilization of maternal health services, it fails to demonstrate the effect of social media usage on the knowledge, attitude and practices of the study subjects.

Action: It is therefore recommended that since the data source did not capture the datasets that appropriately describe contemporary media channels such as social media and internet usage, the authors may wish to review the title of the paper to reflect the source of data.

Response from author: Thank you very much for your valuable observations and possible actions to the mentioned points. This aspect (internet access) is already mentioned in the limitations section as the Demographic Health Survey (DHS) dataset lacked the information on the usage of internet and women’s subsequent social media use. Bangladesh, as a developing nation, not famous for having internet access especially in rural/semi-urban setting among rural population; so, may be that was the reason for the Demographic Health Survey (DHS) not to explore the use of social media in their questionnaire (I do not know, actually). I am sure these aspects will be included in future endeavors so that we can study its effect on the essential health service use. But currently, the study findings are based on the available resources that we have. 

However, the title is revised to “traditional media access” rather than “media access” as per your suggestion mention on the “action:”

“How do traditional media access and mobile phone use affect maternal healthcare service use in Bangladesh? Moderated mediation effects of socioeconomic factors”.

2. Definition of key terms: While the authors defined ‘have some access’ as ‘Watched TV/listened to radio/read newspapers at least once a week’, it is completely silent on the definition of ‘limited access’. It graded the level of access to media use (some access and limited access) with accompanying degree of maternal health service utilization but failed to do same for mobile phone use assuming that ‘any kind of exposure’ to mobile technology use will positively affect maternal health service utilization outcome.

Action: The authors should clearly define ‘limited access’ so as to avoid ambiguity to the target audience.

Response from author: “Limited access” term is defined as per your suggestion. 

Have some access: Watched TV/listened to radio/read newspapers at least once a week’

Limited access: Watched TV/listened to radio/read newspapers less than once a week or not at all.

Regarding mobile use, the dataset didn’t have any variable that categorizes the frequency of using mobile (exposure) for maternal health service use; it was limited to the reasons of using mobile phone. It can be also said that media is something people attached to the more frequent basis (daily/weekly, etc.), but utilizing mobile phone for health service use is something depends on the need basis. It could be due to the fact that if a woman has a mobile, she might be using it every day for general purpose and use it for health service when required. The survey dataset did not particularly segregate how frequently a woman has used mobile for anything related to health service use rather examined how many women have used it for health service. But I understood what you meant. Thank you for this observation. 

3. Oversimplification of findings: The authors’ address a critical topic in demand creation within a complex system of factors affecting uptake of comprehensive maternal care services amongst pregnant mothers. For instance, the influence of counter moderating factors such as increased distance to health facilities on the use of ANC services were completely ignored. Also, In Page 27, paragraph 2, line 8, the authors ignored the influence of cultural beliefs, practices, and norms on the lack of women’s autonomy in making healthcare decisions for themselves in the developing world by propounding that having a cell phone to make enquiry about nearby health services is a solution.

Therefore, this paper does not sufficiently describe the balance of complex demand and supply forces for achieving desirable maternal health services outcomes. 

Action: Providing this information will furrther strengthen the paper

Response from the author:

Thank you for this comment. 

We have included place of residence (rural vs. urban) as an important predictor to minimize distance factor (as a proxy) and tested its effect on the association; it has significant moderating effects on the relationship of media access and number of antenatal care visits. 

Women’s autonomy in health decision making was already included as one of the predictors and its effect was minimized, although its effect was found insignificant. The dataset also lacks information on the influence of cultural beliefs, and practices (a certain qualitative component) on women’s autonomy and it was mentioned in the discussion section as well. 

This study findings did not claim that having access to media and mobile phone is an absolute solution to this maternal health service issue rather can act as a catalyst that can facilitate interpersonal communication and more information provision to the women.

The whole study is based on the available resources (dataset produced by DHS); it’s understandable that the study topic is itself a complex phenomenon that involves factors from different dimensions and we tried to include and test the effect of possible available variables that can influence the association under investigation.

---

## [Decision Letter · Decision Letter 1]

24 Mar 2022

How do traditional media access and mobile phone use affect maternal healthcare service use in Bangladesh? Moderated mediation effects of socioeconomic factors

PONE-D-22-02833R1

Dear Dr. Kabir,

We’re pleased to inform you that your manuscript has now been judged scientifically suitable for publication in PLOS ONE and will be formally accepted for publication once it meets all outstanding technical requirements.

Kind regards,

Bassey E. Ebenso, Ph.D., M.P.H., M.D.,

Academic Editor

PLOS ONE

Reviewers' comments:

Reviewer's Responses to Questions

**Comments to the Author**

1. If the authors have adequately addressed your comments raised in a previous round of review and you feel that this manuscript is now acceptable for publication, you may indicate that here to bypass the “Comments to the Author” section, enter your conflict of interest statement in the “Confidential to Editor” section, and submit your "Accept" recommendation.

Reviewer #1: All comments have been addressed

Reviewer #2: All comments have been addressed

2. Is the manuscript technically sound, and do the data support the conclusions?

Reviewer #1: Yes

Reviewer #2: Yes

3. Has the statistical analysis been performed appropriately and rigorously? 

Reviewer #1: Yes

Reviewer #2: Yes

4. Have the authors made all data underlying the findings in their manuscript fully available?

Reviewer #1: Yes

Reviewer #2: Yes

5. Is the manuscript presented in an intelligible fashion and written in standard English?

Reviewer #1: Yes

Reviewer #2: Yes

6. Review Comments to the Author

Reviewer #1: I am satisfied reading the paper it is now more obvious than previous version. Although, I couldn’t find the prevalence rate.

However, happy working best of luck.

Reviewer #2: The authors have satisfactorily addressed all the issues raised. The manuscript is now acceptable for publication

7. PLOS authors have the option to publish the peer review history of their article (what does this mean?). If published, this will include your full peer review and any attached files.

Reviewer #1: No

Reviewer #2: No

---

## [Editor Report · Acceptance letter]

29 Mar 2022

PONE-D-22-02833R1 

How do traditional media access and mobile phone use affect maternal healthcare service use in Bangladesh? Moderated mediation effects of socioeconomic factors 

Dear Dr. Kabir:

I'm pleased to inform you that your manuscript has been deemed suitable for publication in PLOS ONE. Congratulations! Your manuscript is now with our production department. 

Kind regards, 

on behalf of

Dr. Bassey E. Ebenso 

Academic Editor

PLOS ONE